# Hypopituitarism in *Sox3* null mutants correlates with altered NG2-glia in the median eminence and is influenced by aspirin and gut microbiota

Christophe Galichet[1,2]*, Karine Rizzoti[1]*, Robin Lovell-Badge[1]*

1 Stem Cell Biology and Developmental Genetics Lab, The Francis Crick Institute, London, United Kingdom,
2 Neurobiological Research Facility, UCL Sainsbury Wellcome Centre for Neural Circuits and Behaviour, London, United Kingdom

* c.galichet@ucl.ac.uk (CG); karine.rizzoti@crick.ac.uk (KR); robin.lovell-badge@crick.ac.uk (RLB)

## Abstract

The median eminence (ME), located at the base of the hypothalamus, is an essential centre of information exchange between the brain and the pituitary. We and others previously showed that mutations and duplications affecting the transcription factor *SOX3/Sox3* result in hypopituitarism, and this is likely of hypothalamic origin. We demonstrate here that the absence of *Sox3* predominantly affects the ME with phenotypes that first occur in juvenile animals, despite the embryonic onset of SOX3 expression. In the pituitary, reduction in hormone levels correlates with a lack of endocrine cell maturation. In parallel, ME NG2-glia renewal and oligodendrocytic differentiation potential are affected. We further show that low-dose aspirin treatment, which is known to affect NG2-glia, or changes in gut microbiota, rescue both proliferative defects and hypopituitarism in *Sox3* mutants. Our study highlights a central role of NG2-glia for ME function during a transitional period of post-natal development and indicates their sensitivity to extrinsic signals.

## Author summary

The hypothalamus, a complex structure in the ventral-medial brain, has many critical functions, including controlling the secretion of hormones from the pituitary gland. In turn, these hormones regulate many important physiological functions, such as growth, stress responses and reproduction, and insufficient levels result in significant morbidity and even mortality if untreated. The hypothalamus is linked to the pituitary via the median eminence (ME), where information is exchanged. Previously, we, and others, have highlighted the importance of *SOX3/Sox3* for the hypothalamo-pituitary axis, because mutations in this gene result in a reduction of all pituitary hormones, or hypopituitarism, in both humans and mice. Here we demonstrate that loss of *Sox3* in mice affects specialised supporting cells of the ME called NG2-glia and that this only occurs only after weaning. The defects correlate with abnormal development of the hormone-secreting cells of the pituitary and the onset of hypopituitarism. We also show that external factors

**Data Availability Statement:** Gut microbiota datasets are publicly available at the Sequence Read Archive (SRA) of the National Library of Medicine (PRJNA1113545). All other relevant data

are within the manuscript and its Supporting information files.

**Funding:** This work was supported by the Medical Research Council, U.K. (U117512772 for CG, KR and RLB, U117562207 for CG, KR and RLB and U117570590 for CG, KR and RLB) and by the Francis Crick Institute which receives its core funding from Cancer Research UK (CC2116 for CG, KR and RLB), the UK Medical Research Council (CC2116 for CG, KR and RLB), and the Wellcome Trust (CC2116 for CG, KR and RLB). The funders had no role in study design, data collection and analysis, decision to publish, or preparation of the manuscript.

**Competing interests:** The authors have declared that no competing interests exist.

such as aspirin or changes in gut microbiota can rescue both the hypopituitarism in *Sox3* mutants and renewal of the ME NG2-glia. In conclusion, our data suggest a critical role for NG2 glia in the ME for the maturation of hormone producing cells in the pituitary and that these NG2 glia are sensitive to the loss of *Sox3* and to extrinsic factors. This argues that aspirin, altering gut microbiota, or perhaps just dietary modulation may be of benefit in treating certain cases of hypopituitarism.

## Introduction

The hypothalamus is a crucial regulator of homeostasis and energy balance in vertebrates. It centralizes information from other regions of the brain and from the periphery, while an important part of its function is to regulate the secretion of pituitary hormones. These control essential functions, including post-natal growth, puberty, pregnancy, fertility, lactation, stress and energy homeostasis. Consequently, pituitary hormone deficiencies (hypopituitarism) are associated with significant morbidity.

The murine pituitary gland, which is located under the hypothalamus, has three compartments: the anterior pituitary lobe (AL), containing five different endocrine cell types, the intermediate (IL) with one (melanotrophs), and the posterior pituitary, composed of glial cells and hypothalamic axonal termini. The hypothalamus connects to the pituitary via the pituitary stalk, which itself contains neuronal, glial, and vascular components.

AL endocrine cells are regulated by neurohormones secreted by different hypothalamic neurons into a bed of fenestrated capillaries located at the base of the third ventricle, in a structure called the median eminence (ME). These neurohormones are delivered to the anterior pituitary by the hypophyseal portal system. Conversely, for the posterior pituitary, neurohormones (oxytocin and vasopressin) are released directly from hypothalamic axons that reach the gland itself [1]. The ME, which is located outside the blood-brain barrier [2], and the pituitary stalk are crucial components of the hypothalamo-pituitary axis, as flow of information in and out of the hypothalamus is respectively collected and conveyed through these structures. In addition to the blood vessels, several cell types, including glial cells and microglia, are found in the ME (for review see [2,3]). NG2-glia, or oligodendrocyte precursor cells (OPCs), are present in ME [4–6]. They are the most proliferative cells within the central nervous system (CNS) and their canonical function is to generate oligodendrocytes, although rarely astrocytes and neurons in some contexts. However, they also fulfil other roles, such as interacting with and modulating the function of neighbouring neurons, astrocytes or microglia (for review see [7]). Within the ME, NG2-glia have been shown to generate oligodendrocytes [5,6], some of which are in close contact with the dendrites of leptin receptor expressing neurons. The relevance of these interactions is revealed by ablation of ME NG2-glia which results in the degeneration of these dendrites, leading to increased body weight [4]. Furthermore, refeeding after an overnight fast induces a rapid proliferation and differentiation of ME NG2-glia [6], demonstrating that nutrition influences, and is influenced by, ME NG2-glia. Another category of glial cells, the tanycytes, of which there are several types [8], are flanking the base of the third ventricle. Because the ME is located outside the blood-brain barrier, tanycytes are exposed to both the systemic circulation and the cerebro-spinal fluid [9]. They regulate neurohormone secretion from hypothalamic axon terminals [10–12] and modulate the release of blood-borne peripheral information to the hypothalamus ([13] and for review see [14]). Furthermore, they comprise a population of hypothalamic stem cells (SCs) (for review see [2,15]). Similarly to other CNS stem cell populations, tanycytes express the HMG-box transcription factor SOX2

[16–20], required for maintenance of several different adult stem cell populations (for review see [21]). In embryonic neural progenitors, SOX2 is often co-expressed with the two other SOXB1 members, SOX1 and SOX3 [22]. Upon cell differentiation, either in embryonic or adult contexts, expression of *Soxb1* genes, tends to be downregulated [23]. There are exceptions, however. For example, SOX2 and SOX3 are maintained in spinal cord oligodendrocyte precursors, where they are required for their subsequent differentiation [24].

In both humans and mice, mutations in *SOX3/Sox3*, which is located on the X chromosome, are associated with hypopituitarism (for review see [25]). In mice, we showed that SOX3 is largely absent from the pituitary, but it is found in the developing and mature hypothalamus, being particularly highly expressed in the embryonic infundibulum and later maintained in the ME ([26] and this study). Here, we aimed to characterise the role of SOX3 in the hypothalamus and the etiology of the hypopituitarism. Using conditional gene deletion, we show that loss of *Sox3* in the central nervous system (CNS) is sufficient to cause hypopituitarism and that this develops postnatally, at the time of weaning. Consequently, we observe in the pituitary a defective maturation of endocrine cells. SOX3 is expressed in hypothalamic stem cells and its deletion affects their proliferation *in vivo* and in hypothalamic neurosphere assays. SOX3 is also expressed in NG2-glia in the ME, and both their proliferation and differentiation are reduced in *Sox3* mutants. Furthermore, we observe a striking deficiency in myelination of axons in the ME. Importantly, the appearance of all these defects correlates with development of hypopituitarism suggesting that SOX3 is required postnatally for the functional maturation of the ME. Building on previous work demonstrating the efficacy of aspirin in inducing proliferation and/or differentiation of NG2-glia [27,28], we treated *Sox3* mutants with low dose aspirin. We observed a restoration of NG2-glia proliferation, and a rescue of pituitary hormonal deficiencies, but not a rescue of myelination. We also found that changes in the gut microbiome ameliorate the hypopituitarism. Our study highlights a novel role for NG2-glia in the post-weaning formation of a functional ME and its control of pituitary maturation, and that an otherwise robust phenotype due to loss of *Sox3* in these cells can be modified by non-genetic factors.

## Results

### 1. SOX3 is required in the hypothalamus for post-natal pituitary endocrine cell maturation

SOX3 is widely expressed in the developing CNS [22], but it is also retained within some brain regions postnatally, notably the hypothalamus [26], where its expression is found in tanycytic neural stem cells (NSCs) and several differentiated cell types [29]. In contrast, we only observed SOX3 in rare cells within the pituitary representing a very small subset of lactotrophs (S1A Fig). Therefore, the hypopituitarism displayed by *Sox3* mutants is likely to be of hypothalamic origin.

To test this hypothesis, we used *Nestin-Cre* [30] and *Pou1f1-Cre* [31] to delete *Sox3* in the CNS and the pituitary respectively. Pituitary hormonal contents were then measured by radio immunoassay (RIA) in two-month-old males (Fig 1A). While *Nestin*-Cre animals display a modest but significant reduction in growth hormone (GH) levels, as we previously reported [32], we observed a significant reduction of GH contents in *Nestin*-Cre*; Sox3*<sup>floxGFP/Y</sup> mice compared to both *Nestin*-Cre and *Sox3*<sup>floxGFP/Y</sup> controls. This 75% reduction in *Nestin*-Cre*; Sox3*<sup>floxGFP/Y</sup> mice is comparable to that measured in *Sox3*<sup>-/Y</sup> mutant males (Fig 1A, [26]). In contrast, in *Pou1f1*-Cre*; Sox3*<sup>floxGFP/Y</sup> mice, neither GH (Fig 1A) nor Prolactin (PRL) (S1B Fig) contents are affected. These data are therefore in agreement with a CNS origin of the

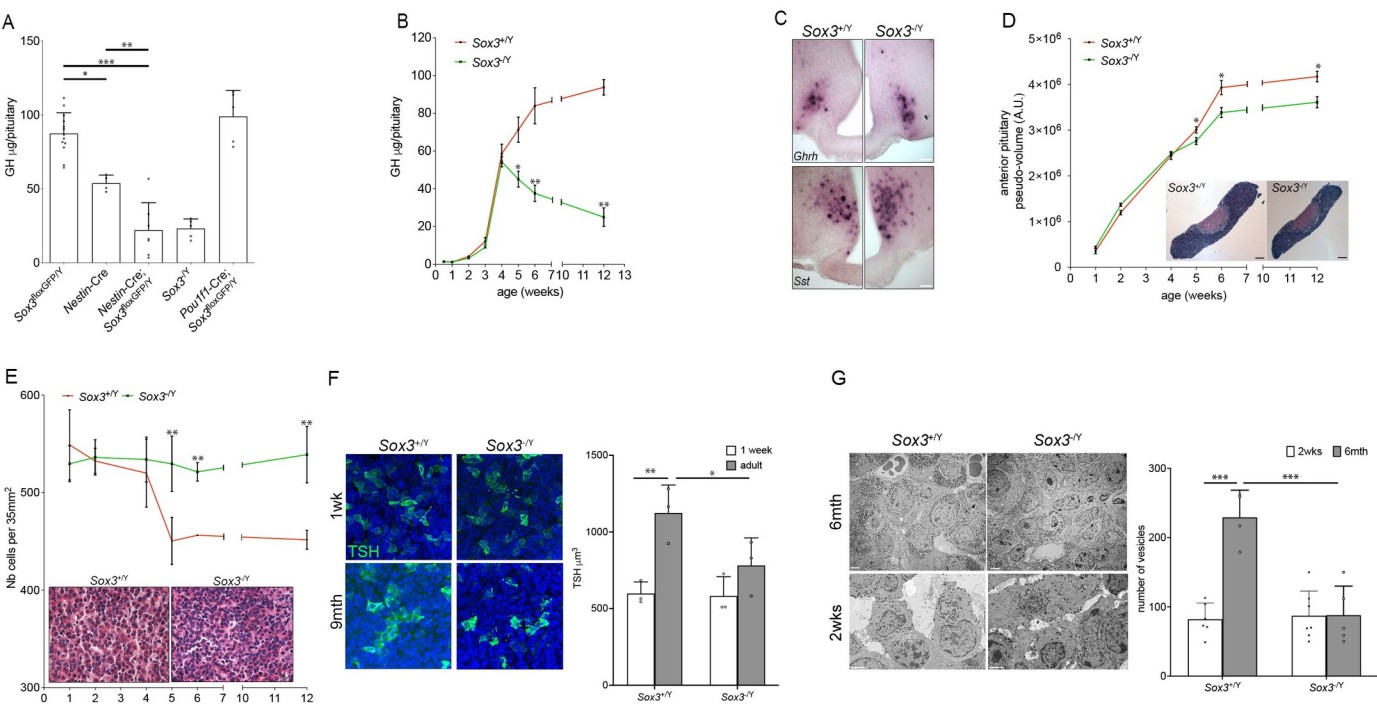

**Fig 1. The hypopituitarism in *Sox3*$^{-/Y}$ has a hypothalamic origin and develops after weaning, concomitantly with defective pituitary endocrine cell maturation.** (A) GH contents of 2-month *Sox3*$^{floxGFP/Y}$, *Nestin*-Cre, *Nestin*-Cre; *Sox3*$^{floxGFP/Y}$, *Sox3*$^{-/Y}$ and *Pit1*-Cre; *Sox3*$^{floxGFP/Y}$ pituitaries. Reduction in GH contents only appears when SOX3 is removed from the CNS. (B) GH contents of *Sox3*$^{+/Y}$ and *Sox3*$^{-/Y}$ at different post-natal weeks. Reduction in GH contents in *Sox3*$^{-/Y}$ animals appears post-weaning. (C) *In situ* hybridization for *Ghrh* (upper panels) and *somatostatin* (*Sst*, lower panels) on 2-month *Sox3*$^{+/Y}$ and *Sox3*$^{-/Y}$ hypothalami. Scale bar: 100μm. (D) *Sox3*$^{+/Y}$ and *Sox3*$^{-/Y}$ anterior pituitary sizes at different post-natal weeks. Insets show 2-month *Sox3*$^{+/Y}$ and *Sox3*$^{-/Y}$ H&E-stained sections where the anterior pituitary is outlined. *Sox3*$^{-/Y}$ anterior pituitaries are smaller post-weaning. Scale bar: 200μm. (E) *Sox3*$^{+/Y}$ and *Sox3*$^{-/Y}$ anterior pituitary cell densities. Insets represent 2-month *Sox3*$^{+/Y}$ and *Sox3*$^{-/Y}$ H&E-stained anterior pituitaries. *Sox3*$^{-/Y}$ anterior pituitary cell densities remain constant while these decrease in *Sox3*$^{+/Y}$ post-weaning. Scale bar: 15μm. (F) *Sox3*$^{+/Y}$ and *Sox3*$^{-/Y}$ TSH+ve cell volumes at 1 week (white columns) and 2 months (grey columns). Insets show 1-week and 9-month *Sox3*$^{+/Y}$ and *Sox3*$^{-/Y}$ pituitaries immunolabelled for TSH. TSH+ve cell volumes increase with age in *Sox3*$^{+/Y}$ but remain constant in *Sox3*$^{-/Y}$ mice. (G) Transmission electron microscopy images of 2-week and 6-month *Sox3*$^{+/Y}$ and *Sox3*$^{-/Y}$ anterior pituitaries. The bar chart shows the number of secretory granules in *Sox3*$^{+/Y}$ and *Sox3*$^{-/Y}$ pituitaries at 2 weeks (white columns) and 6 months (grey columns). Note that fewer secretory granules are present in 6-month *Sox3*$^{-/Y}$ anterior pituitaries. Scale bar: 2μm. *: $p < 0.05$; **: $p < 0.01$; ***: $p < 0.001$.

hypopituitarism displayed by *Sox3*$^{-/Y}$ animals. To avoid undesirable effects originating from the *Nestin*-Cre transgene, we pursued our subsequent analyses on *Sox3*$^{-/Y}$ animals.

We then measured GH contents at different time points after birth to determine when deficits appear in *Sox3*$^{-/Y}$ males (Fig 1B). After birth, GH levels rise normally in mutants compared to *Sox3*$^{+/Y}$ animals, but decline sharply shortly after weaning, down to 25% of the levels observed in *Sox3*$^{+/Y}$ animals at two months of age. Similar post-weaning deficiencies were measured for PRL, thyroid-stimulating hormone (TSH) and luteinizing hormone (LH) (S1C Fig; [26]) and similar results were obtained when analysing *Gh* and *Prl* gene expression (S1D Fig). Because the growth spurt observed in juveniles is known to be regulated by changes in growth hormone releasing hormone (GHRH) pulsatility [33], this result is again in accord with a hypothalamic origin of the phenotype in *Sox3*$^{-/Y}$ animals.

We did not observe any obvious morphological defect in the hypothalami in *Sox3*$^{-/Y}$ adult males [26]. We thus examined in more detail the arcuate and paraventricular nuclei, both involved in GH secretion control, by performing *in situ* hybridisation using *Ghrh* and *somatostatin* (*Sst*) mRNA probes (Fig 1C). Their expression patterns are similar in *Sox3*$^{+/Y}$ and *Sox3*$^{-/}$

[Y] animals, again indicating that cell specification and patterning within the hypothalamus are unaffected.

To understand which aspect of hypothalamic function is affected in mutants, we examined the consequences of *Sox3* loss on the pituitary. In parallel with the appearance of the hormonal deficiencies, we observed that the growth of the anterior lobe slows down significantly in *Sox3*[-/Y] compared to *Sox3*[+/Y] mice after weaning (Fig 1D). Furthermore, while the cellular density in *Sox3*[+/Y] animals sharply decreases between 4 and 5 weeks of age, it remains constant in *Sox3*[-/Y] (Fig 1E). This suggests that endocrine cells, whose secretory capacities usually increase dramatically postnatally [34], remain small in *Sox3* mutants. To confirm this hypothesis, we measured the volume of TSH secreting cells. We chose these because they represent a small population, about 5% of AL cells [35], and because they are individualized, which thus make them suitable for cell volume measurement. In *Sox3*[+/Y] animals, TSH positive cells double their volume from one-week old to adulthood. In contrast, in *Sox3* mutants, their volume does not significantly increase during the same period (Fig 1F). This suggests that hormonal secretory capacity/maturation is affected in mutants. We directly examined this aspect by performing transmission electron microscopy (TEM) and observed a striking reduction in the number of secretory granules in cells of *Sox3*[-/Y] ALs (Fig 1G), implying impaired endocrine cell terminal differentiation. Altogether these data show that the hypopituitarism displayed by *Sox3*[-/Y] mice is of hypothalamic origin. The consequence, at the pituitary level, is an impairment of endocrine cell maturation, which becomes apparent after weaning and results in the production of insufficient levels of hormones.

## 2. Hypothalamic progenitor maintenance *in vitro* relies on SOX3

To determine the origin of the hypothalamic defect, we analysed SOX3 expression in more detail (Fig 2A). SOX3 is expressed in cells lining the third ventricle and in scattered cells in the surrounding parenchyma. In the latter, it is present in some neurons (NeuN, Figs 2B and S2A), as previously shown [29], and in a few astrocytes (GFAP, Figs 2C and S2A) and NG2-glia (PDGFRa, Figs 2E–2F and S2A). SOX3 is also present in Nestin[+ve] tanycytes (Fig 2D). Beside these 3 cell types, it is likely that SOX3 is alos expressed in neuroblasts and during oligodendrocyte differentiation, as shown in the spinal cord [24]. Within the ME, SOX3 is found in NG2-glia (PDGFRa, Fig 2E; NG2, S2B Fig), but not in MAG[+ve] oligodendrocytes (S2C Fig).

Because SOX3 is expressed in hypothalamic tancyte NSCs, and SOXB1 proteins are known to be required for NSC emergence and maintenance (for review see [21]), we first performed *in situ* hybridization for the tanycyte marker *Rax* [36]. *Rax* expression pattern between *Sox3*[-/Y] and *Sox3*[+/Y] animals is similar, albeit possibly slightly reduced in *Sox3*[-/Y] hypothalmi, suggesting that deletion of *Sox3* does not affect tanycyte emergence (Fig 2G). We then investigated neurosphere forming ability of hypothalamic progenitors from 6-week-old animals [37] (Fig 2H). We generated *Sox3*[+/Y] hypothalamic neurospheres which maintain expression of SOX3 (Fig 2Hiii), as previously shown for embryonic ones [38]. Neurospheres from *Sox3*[-/Y] hypothalamic progenitors were formed and these express GFP, the open reading frame of which replaces that of *Sox3* (Fig 2Hiv) and not SOX3 (S3A–S3B Fig) [26]. There was initially no difference in the number of plated cells and of primary neurospheres formed between *Sox3*[+/Y] and *Sox3*[-/Y] animals. However, by the third passage, the total number of cells and size of *Sox3*[-/Y] spheres had declined sharply (Fig 2H–2I–red lines). In subsequent passages, most *Sox3*[-/Y] cells adhere and very few neurospheres are formed, with none arising by the 9[th] passage. We then analysed cell proliferation and apoptosis on adherent NSC cultures of dissociated neurospheres [39]. There were significantly fewer phospho-histone H3[+ve] mitotic cells in *Sox3*[-/Y] samples (Fig 2J) while no increase in cell apoptosis was observed by TUNEL assay (S4A Fig).

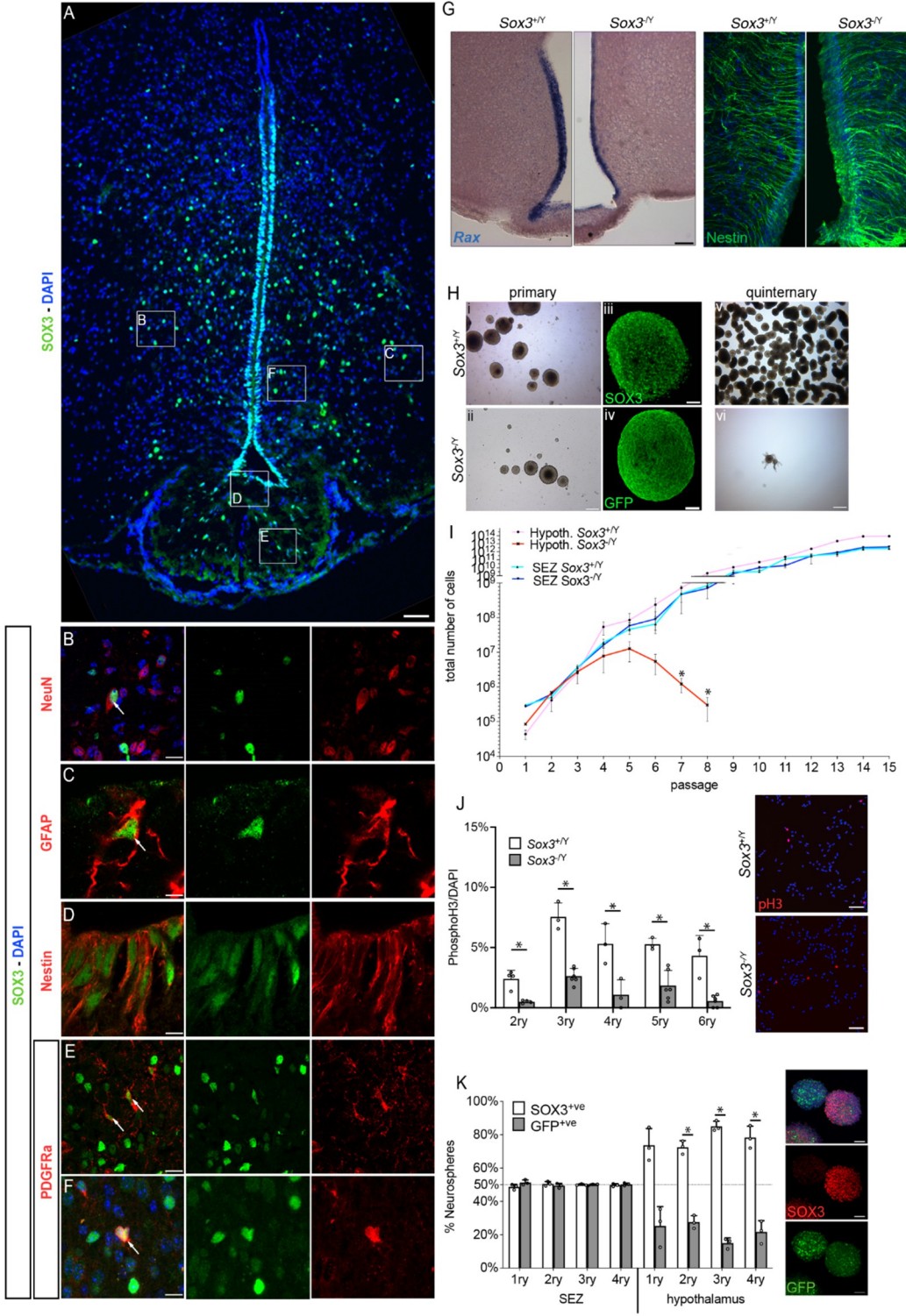

**Fig 2. SOX3 is expressed in hypothalamic SCs and NG2-glia, and it is required for hypothalamic neurosphere maintenance.** (A- F) Fluorescent immunolabelling for SOX3 alone (A) or together with NeuN (Neurons—B), GFAP (astrocytes—C), Nestin (Tanycyctes—D) or PDGFRa (NG2-glia—E-F) in 2 month-old hypothalami. Arrows point to double positive cells. Scale bar: 100µm (A), 10µm (C, F), 20µm (B, D, E). (G) *In situ* hybridization for *Rax* and fluorescent immunolabelling for Nestin on 2-month *Sox3*^+/Y^ and *Sox3*^-/Y^ hypothalami. Scale bar: 100µm. (H) *Sox3*^+/Y^ (i, iii, vi) and *Sox3*^-/Y^ (ii, iv, vii) hypothalamic derived neurospheres at primary (i,ii) or quintenary (vi,vii) passage. Scale bar: 200µm

(primary) and 400μm (quintenary). Insets show neurospheres immunolabelled for SOX3 (*Sox3*<sup>+/Y</sup> spheres—iii) or GFP (*Sox3*<sup>-/Y</sup> spheres—iv). Scale bar: 50μm. (I) Total number of cells after each passage of hypothalamic derived (red lines) or SEZ derived (blue lines) neural stem cells from *Sox3*<sup>+/Y</sup> (light colour) or *Sox3*<sup>-/Y</sup> (dark colour) mice. *Sox3*<sup>-/Y</sup> hypothalamic derived neurospheres cannot be maintained in culture. (J) Percentage of mitotic cells in progenitor monolayer culture. Insets represent mitotic cells immunolabelled with phospho-Histone 3. Scale bar: 50 μm. There is a proliferation defect in *Sox3*<sup>-/Y</sup> progenitors. (K) Percentage of *Sox3*<sup>+/ΔGFP</sup> neurospheres containing mostly SOX3<sup>+ve</sup> cells (white columns) or mostly GFP<sup>+ve</sup> cells (grey columns) derived from SEZ or hypothalami. *Sox3*<sup>+/ΔGFP</sup> hypothalamic neurospheres are not maintained in culture. *: p<0.05.

In contrast, formation and maintenance of neurospheres from the sub-ependymal zone (SEZ) of the lateral ventricles, a well characterised adult neurogenic niche [40], which also express SOX3 [41], is normal in adult *Sox3*<sup>-/Y</sup> animals (Fig 2I–blue lines). Furthermore, hypothalamic neurospheres made from *Sox3*<sup>-/Y</sup> mice before the hypopituitarism appears can also be propagated and proliferate normally (S5A and S5B Fig). We analysed the differentiation potential of pre and post-weaning hypothalamic NSCs *in vitro* (S6A and S6B Fig). As previous shown [19,38], we observed differentiation into the three CNS lineages, βIII-tubulin<sup>+ve</sup> neurons, GFAP<sup>+ve</sup> astrocytes and CNPase<sup>+ve</sup> oligodendrocytes in *Sox3*<sup>+/Y</sup> and *Sox3*<sup>-/Y</sup> animals. Therefore, SOX3 is necessary for self-renewal, but not differentiation *in vitro*, in hypothalamic neurospheres and this requirement correlates in time with the onset of hypopituitarism in mutant animals.

We next performed neurosphere assays using *Sox3*<sup>+/ΔGFP</sup> females (Fig 2K). Due to the location of *Sox3* on the X-chromosome and to random X chromosome inactivation, cells from these mice that normally transcribe *Sox3* should express either SOX3 or GFP in roughly equal proportions. This was seen in neurospheres derived from the SEZ of the lateral ventricles of two-month- or ten-day old *Sox3*<sup>+/ΔGFP</sup> animals and in spheres derived from the hypothalamus of ten-day old *Sox3*<sup>+/ΔGFP</sup> pups cultured in non-clonal conditions (Figs 2K and S5C). In contrast, neurospheres from the hypothalamus of two-month-old *Sox3*<sup>+/ΔGFP</sup> animals, show a significant reduction in the contribution of GFP<sup>+ve</sup>, SOX3<sup>-ve</sup> cells, compared to SOX3<sup>+ve</sup>, GFP<sup>-ve</sup> cells (Fig 2K). This mosaic analysis confirms and expands our previous analyses, showing that SOX3 is required cell-autonomously for hypothalamic tanycyte NSC self-renewal *in vitro*, again, in an age-dependent manner.

## 3. SOX3 is required in the ME for cell renewal

We then examined cell proliferation *in vivo* by performing BrdU incorporation experiments. In *Sox3*<sup>+/Y</sup> adult hypothalami, we observed BrdU<sup>+ve</sup> cells within the ME and the parenchyma (Fig 3B), as previously shown (for review see [14,42]). In both *Sox3*<sup>+/Y</sup> and *Sox3*<sup>-/Y</sup> adult hypothalami, BrdU<sup>+ve</sup> cells are mostly located within the body of the ME rather than in ventricle lining cells (Fig 3B). We observed fewer BrdU<sup>+ve</sup> cells in *Sox3*<sup>-/Y</sup> versus *Sox3*<sup>+/Y</sup> animals (Fig 3B–3C), predominantly affecting the ME where a 40% reduction in BrdU incorporation/ ME surface area is observed in *Sox3*<sup>-/Y</sup> (Fig 3D). ME is also significantly smaller in mutants (Fig 3D). Because we did not observe any increase in the number of apoptotic cells in *Sox3*<sup>-/Y</sup> mutants compared to *Sox3*<sup>+/Y</sup> mice (S4B Fig), the smaller size of the ME is likely explained by its reduced proliferative capacity. In contrast, BrdU incorporation in the dentate gyrus, another well studied neurogenic niche [40], of 2-month-old *Sox3*<sup>-/Y</sup> is similar to that of *Sox3*<sup>+/Y</sup> (Fig 3C). There was also no difference in the number of BrdU<sup>+ve</sup> cells in 1-week-old hypothalami between *Sox3*<sup>+/Y</sup> and *Sox3*<sup>-/Y</sup> littermates, i.e. before the hypopituitarism appears in the latter (S5D Fig).

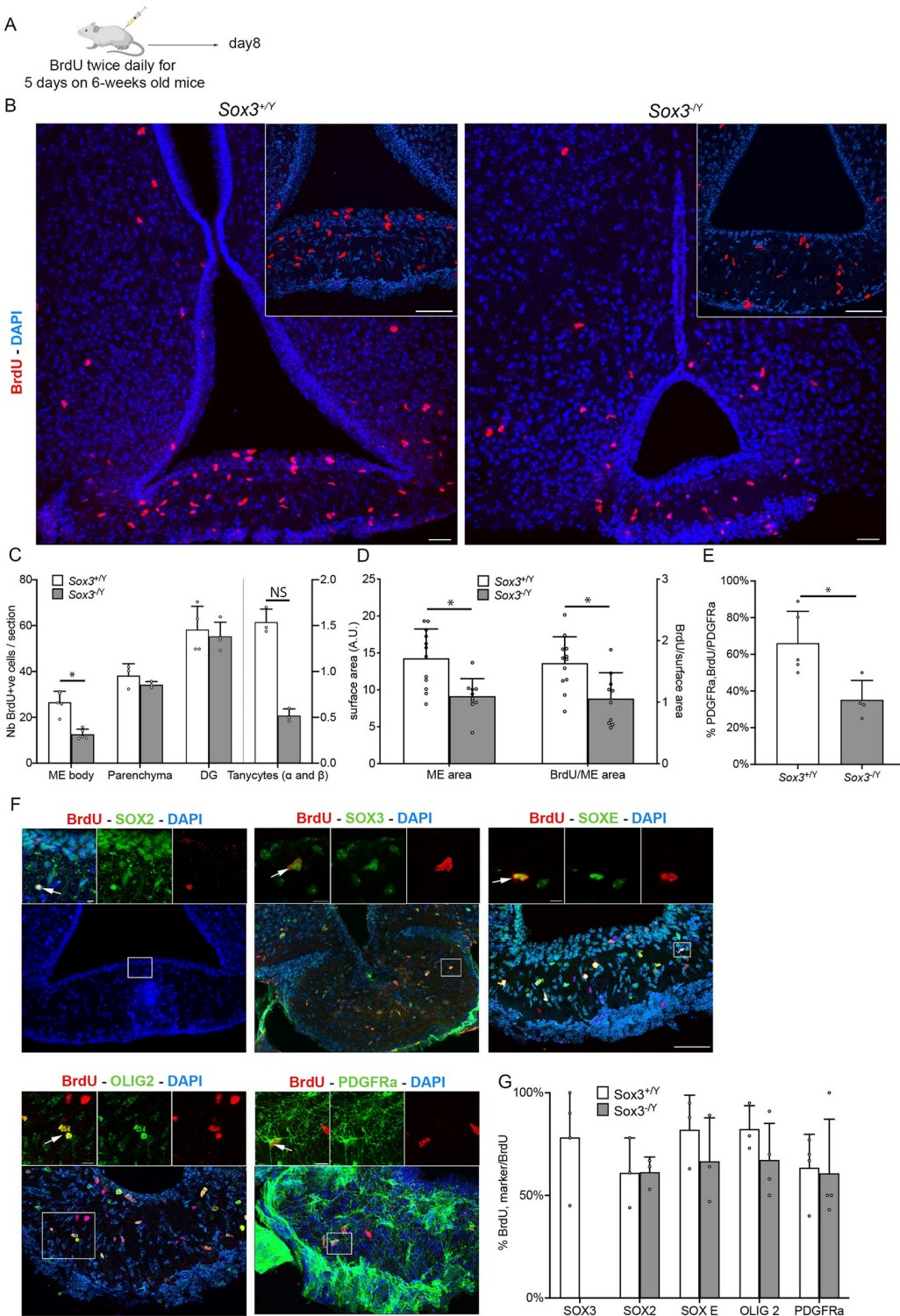

**Fig 3. Cell renewal in ME is reduced in absence of SOX3.** (A) BrdU administration paradigm. Created with BioRender. com (B) Fluorescent immunolabelling for BrdU on 2-month *Sox3*$^{+/Y}$ and *Sox3*$^{-/Y}$ hypothalami. Insets show ME only. Fewer BrdU$^{+ve}$ cells are present in the *Sox3*$^{-/Y}$ median eminences. Scale bar: 50 μm. (C) Number of BrdU$^{+ve}$ cells in *Sox3*$^{+/Y}$ (white columns) and *Sox3*$^{-/Y}$ (grey columns) median eminences, hypothalamic parenchyma, tanycytes (α and β) and dentate gyri. Fewer BrdU$^{+ve}$ cells are seen in *Sox3*$^{-/Y}$ median eminences. (D) Median eminence area and BrdU density in *Sox3*$^{+/Y}$ (white columns) and *Sox3*$^{-/Y}$ (grey columns) hypothalami. ME are smaller in *Sox3*$^{-/Y}$ mice. (E) Percentage of

BrdU$^{+ve}$, PDGFRa$^{+ve}$ NG2-glia in *Sox3*$^{+/Y}$ (white columns) and *Sox3*$^{-/Y}$ (grey columns) median eminences. (F) Fluorescent immunolabelling for BrdU together with SOX2, SOX3, SOX E, OLIG2 or PDGFRa on 2-month *Sox3*$^{+/Y}$ hypothalami. Scale bar: 50 μm. Insets and arrows point to double-positive cells. Scale bar: 10 μm (G) Percentage of BrdU$^{+ve}$ cells expressing either SOX3, SOX2, SOX E, OLIG2, or PDGFRa, in 2-month *Sox3*$^{+/Y}$ (white columns) and *Sox3*$^{-/Y}$ (grey columns) hypothalami. No difference in the identity of BrdU$^{+ve}$ cells between *Sox3*$^{+/Y}$ and *Sox3*$^{-/Y}$ animals is observed. NS: not significant; *: $p < 0.05$.

Immunochemistry was used to determine the identity of the proliferating cells in the ME. In *Sox3*$^{+/Y}$ animals, the vast majority of BrdU$^{+ve}$ cells express SOX3, SOX2 and one or more of the SOXE proteins SOX8, SOX9 and SOX10 (80, 75 and 82% of BrdU$^{+ve}$ cells respectively, Fig 3F–3G). These cells also express NG2-glia markers, such as PDGFRa and OLIG2 (65 and 80% of BrdU$^{+ve}$ cells respectively Fig 3F–3G), in agreement with studies showing that NG2-glia are the most proliferative cell type in the ME [4,43]. In *Sox3*$^{-/Y}$ mutants, the remaining BrdU$^{+ve}$ cells also represent NG2-glia (Fig 3G), although their proliferation is significantly affected in the ME (Fig 3E).

In summary, SOX3 is required for cell renewal within the ME. Within this structure, NG2-glia proliferation is particularly sensitive to *Sox3* deletion. Moreover, in agreement with an impairment of neurosphere maintenance (Fig 2), proliferation of tanycytes is also affected in mutants (Fig 3C). The reduction in ME cell renewal correlates with the onset of hypopituitarism in *Sox3*$^{-/Y}$ mutants.

## 4. SOX3 supports ME oligodendrocyte differentiation

Because NG2-glia represent the ME cell population mostly affected by *Sox3* deletion, we next characterised their differentiation potential in mutants by performing lineage tracing in *Pdgfra*-CreERT2; *Rosa26*$^{ReYFP}$ animals. In six-week-old *Sox3*$^{+/Y}$ mice, the majority of eYFP$^{+ve}$ cells detected four weeks after tamoxifen induction are PDGFRa and OLIG2 positive NG2-glia, both in the hypothalamic parenchyma and ME (Fig 4B–4C). In the latter, 20% of the eYFP$^{+ve}$ cells have become MAG$^{+ve}$ mature oligodendrocytes. In contrast, in *Sox3*$^{-/Y}$, we observed a significant reduction in the percentage of MAG$^{+ve}$ oligodendrocytes in the progeny of PDGFRa$^{+ve}$ precursors compared to *Sox3*$^{+/Y}$ animals, while there is an increase in the proportion of eYFP; PDGFRa-double positive cells in ME (Fig 4B–4C). These results suggests that ME PDGFRa$^{+ve}$ progenitors are unable to differentiate into oligodendrocytes in the absence of SOX3. We thus examined these cells in the ME. We found both fewer PDGFRa$^{+ve}$ progenitors and mature MAG$^{+ve}$ oligodendrocytes in 2-month-old *Sox3*$^{-/Y}$ ME compared to *Sox3*$^{+/Y}$ ME (Fig 4D). The reduction in PDGFRa$^{+ve}$ cell numbers is in agreement with a reduction in proliferation (Fig 3C), while the decrease in MAG$^{+ve}$ oligodendrocytes fits with an impaired differentiation capacity. We also noticed an increase in cells expressing APC, which is not expressed by NG2-glia, but is a marker of both differentiating and differentiated oligodendrocytes [5]. This indicates that immature oligodendrocytes accumulate in the mutant ME, reflecting a role for SOX3 post weaning in both early and late stages of this lineage (Fig 4D). To confirm the latter, we analysed myelination by transmission electron microscopy (TEM) (Fig 4E). Myelinated axons are clearly visible in *Sox3*$^{+/Y}$ hypothalamic parenchyma and ME, and in the parenchyma of *Sox3*$^{-/Y}$ mutants, however we could not observe any in the mutant ME, in agreement with defective oligodendrocytic differentiation. Because most ME myelinated axons are those of oxytocin and vasopressin (AVP) neurons travelling to the posterior pituitary [44], we analysed potential consequences of defects affecting these. We did not observe any obvious difference in posterior pituitary volume or in the hypothalamic expression of *Avp* in *Sox3*$^{-/Y}$ males (S7A and S7B Fig). AVP deficiency can result in cranial diabetes insipidus, which is characterised by

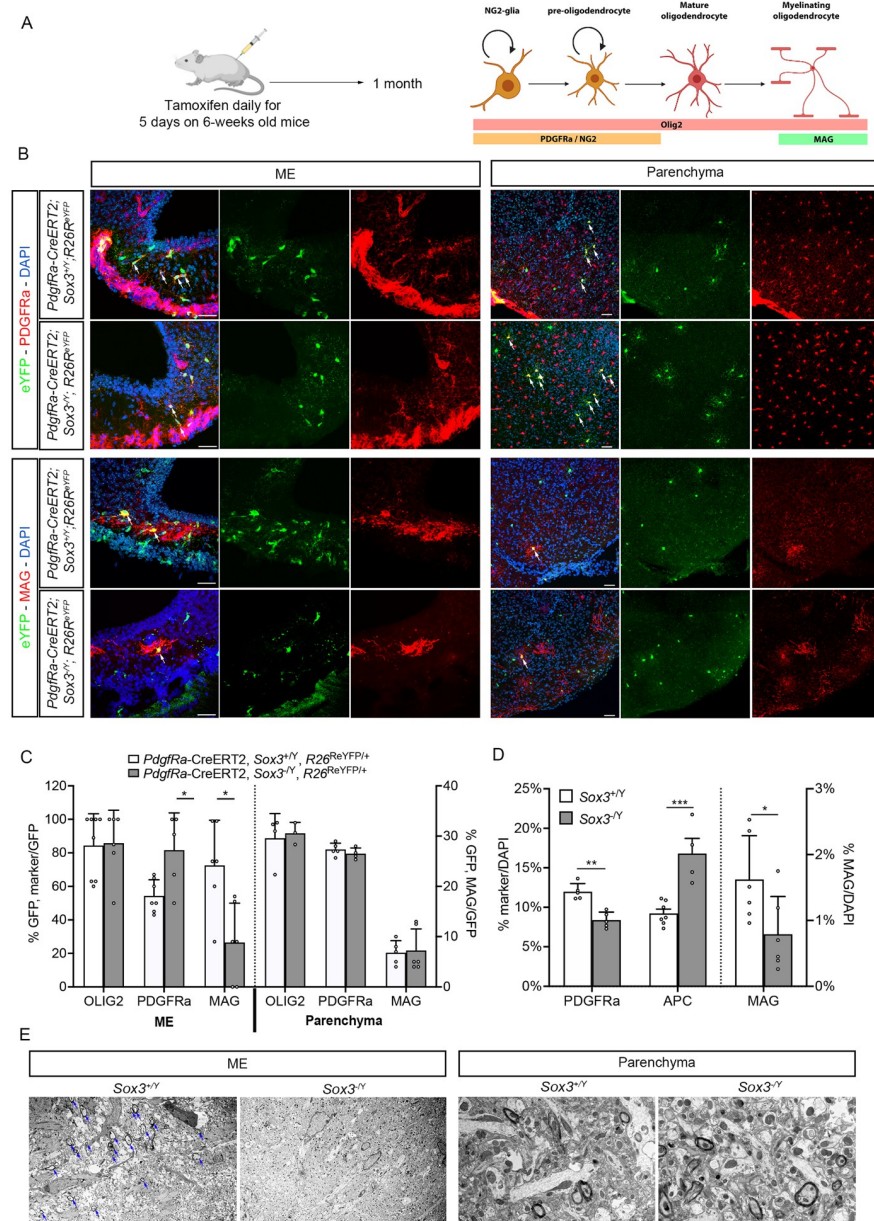

**Fig 4. SOX3 loss results in impaired oligodendrocyte differentiation and myelination in ME.** (A) Injection paradigm and oligodendrogenesis markers used. Created with BioRender.com (B) Fluorescent Immunolabelling for eYFP and PDGFRa (top panels), MAG (bottom panels) on *Pdgfra*-CreERT2*; Rosa26*^ReYFP/+^*; Sox3*^+/Y^ or *Pdgfra*-CreERT2*; Rosa26*^ReYFP/+^*; Sox3*^-/Y^ hypothalami. Left panels show the ME and the right ones hypothalamic parenchyma. Arrows point to double-positive cells. Scale bar: 50 μm. (C) Percentage of recombinant cells expressing OLIG2, PDGFRa or MAG in ME and parenchyma in *Pdgfra*-CreERT2*; Rosa26*^ReYFP/+^*; Sox3*^+/Y^ (white columns) and *Pdgfra*-CreERT2*; Rosa26*^ReYFP/+^*;Sox3*^-/Y^ (grey columns) hypothalami. Fewer PDGFRa^+ve^ progenitors differentiate into MAG positive cells in *Sox3*^-/Y^ MEs. (D) Percentage of cells expressing PDGFRa, APC (left Y axis) or MAG (right Y axis) in *Sox3*^+/Y^ (white columns) and *Sox3*^-/Y^ (grey columns) MEs. There are fewer PDGFRa and MAG present in *Sox3*^-/Y^ MEs. (E) Transmission electron microscopy images of 2-month *Sox3*^+/Y^ or *Sox3*^-/Y^ MEs (left panels) and hypothalamic parenchymas (right panels). Arrows point to myelinated axons in *Sox3*^+/Y^ ME samples. These are absent in *Sox3* mutant MEs. Scale bars: 2μm for ME and 1μm for Parenchyma. *: p<0.05; **: p<0.01; ***: p<0.001.

an increase in water consumption and urine volume, which is hypotonic [45]. We did not find any difference in urine osmalarity between *Sox3*⁺/Y and *Sox3*⁻/Y animals (S7C Fig) suggesting that the AVP axis is not affected in *Sox3*⁻/Y mice. Collectively, this data suggests that the reduced number of NG2-glia and/or the lack of myelination underlies the reduction in pituitary hormone levels in *Sox3*⁻/Y mice.

## 5. Aspirin treatment rescues hypothalamic and pituitary defects in *Sox3* mutants

Two reports have suggested that aspirin can specifically affect NG2-glia [27,28]. In one, both proliferation and differentiation of NG2-glia were induced by low dose aspirin administration *in vitro* and *in vivo* following an ischemic injury in rats [28], while the second suggests that differentiation was exclusively enhanced in neonates following treatment [27]. We therefore decided to examine the effects of a low dose aspirin treatment (12mg/day/kg for 21 days) in 2-month old *Sox3*⁻/Y animals. We first examined BrdU incorporation within ME and observed a significant rescue of the proliferation defect, with the number of BrdU⁺ᵛᵉ cells/section being now similar between *Sox3*⁺/Y and aspirin treated *Sox3*⁻/Y animals (Fig 5A–5C). We then examined the composition of the ME with respect to the oligodendrocyte lineage. We observed a significant increase of PDGFRa⁺ᵛᵉ cells, and a decrease of APC⁺ᵛᵉ cells in aspirin treated compared to untreated *Sox3*⁻/Y mice (Fig 5B–5D). However, we did not observe any significant

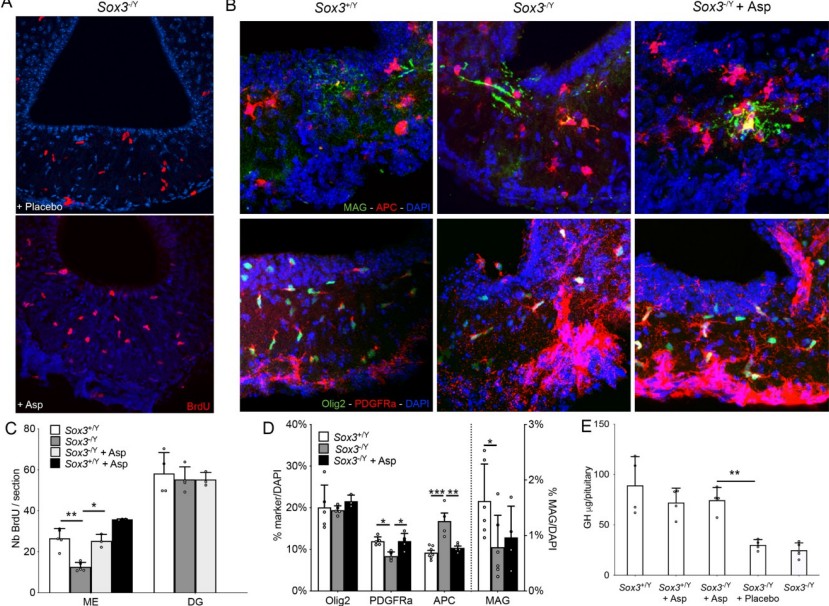

**Fig 5. *Sox3*⁻/Y NG2 proliferation defect and GH deficiencies are rescued by low dose aspirin treatment.** (A-B) Fluorescent immunolebelling for BrdU on 2-month hypothalami of *Sox3*⁻/Y mice receiving placebo pellets (top panel) or low dose aspirin pellets (bottom panel) (A) or for OLIG2, PDGFRa and MAG of *Sox3*⁺/Y, *Sox3*⁻/Y or *Sox3*⁻/Y receiving low dose aspirin (B). Scale bar: 50 μm for A and 20 μm for B. (C) Number of BrdU⁺ᵛᵉ cells in the median eminence or dentate gryus of *Sox3*⁺/Y (white columns), *Sox3*⁻/Y (dark grey columns), *Sox3*⁻/Y with low dose aspirin (light grey columns) or *Sox3*⁺/Y with low dose aspirin (black columns) mice. The number of BrdU⁺ᵛᵉ cells is rescued in *Sox3*⁻/Y mice treated with low dose aspirin. (D) Composition of the median eminence in regard to the oligodendrocyte lineage using OLIG2, PDGFRa, APC and MAG of *Sox3*⁺/Y (white columns), *Sox3*⁻/Y (grey columns), *Sox3*⁻/Y with low dose aspirin (black columns) mice. (E) GH contents in 2-month *Sox3*⁺/Y, *Sox3*⁺/Y with low dose aspirin, *Sox3*⁻/Y with low dose aspirin, *Sox3*⁻/Y with placebo and *Sox3*⁻/Y pituitaries. GH contents are rescued in *Sox3*⁻/Y mice receiving low dose aspirin. *: $p < 0.05$, **: $p < 0.01$, ***: $p < 0.001$.

increase in MAG[+ve] cells or myelination of axons in the ME with aspirin treatment (Figs 5D and S8A). Finally, we examined pituitary GH (Fig 5E) and ACTH (S8B Fig) levels and observed a clear and significant rescue of the deficiencies in aspirin treated *Sox3*[-/Y] mice. These data show that aspirin is able to restore proliferation of NG2-glia and pituitary function, but that the latter does not depend on differentiation to mature oligodendrocytes or myelination of axons.

## 6. Pituitary function in *Sox3* mutants is sensitive to composition of gut microbiota

Following our relocation and the rederivation by embryo transfer of our animals to the Francis Crick institute (Crick) from the National Institute for Medical Research (NIMR), the hypopituitarism was no longer seen in rederived *Sox3*[-/Y] mice (Fig 6D), while the cranio-facial defects [46] were still present. There were several differences between the new mouse facility and the previous one, ranging from water to cage system, but notably these included the adoption of a free-from animal protein and fish meal diet (Labdiet 5021 and Envigo 2018S at NIMR and Crick respectively—S9A Fig). Because an increasing number of studies have highlighted the relationship between diet and gut microbiome and, in turn, gut microbiome and hypothalamic functions (for review see [47–49]), we compared the gut microbiota of mice hosted in our previous and current institutions. We analysed 4 animals for each group and found substantial differences in gut microbiota composition (Figs 6A, 6B and S9B). We noticed that Bacteroidetes were more abundant at NIMR, while Firmicutes were prominent at the Crick (Fig 6A and 6C). At the species level and using Shannon index which evaluates both abundance and richness of species, we found that the alpha diversity was reduced at the Crick compared to NIMR (Fig 6D). To test whether changes in gut microbiota underlie rescue of hormonal deficiencies, we performed faecal transplant (FT) from NIMR animals into Crick animals, while maintaining the mice on Crick diet (Envigo 2018S). Transplants were successful, with FT mice displaying phylum abundance, log F/B ratio and Shannon index similar to that of NIMR mice (Fig 6A–6D). Notably this was accompanied by reappearance of the hypopituitarism, with *Gh* expression being reduced in *Sox3*[-/Y] FT animals compared to *Sox3*[+/Y] FT animals (Fig 6E). We then analysed the oligodendrocyte lineage, particularly in the ME, of NIMR, Crick and FT *Sox3*[+/Y] and *Sox3*[-/Y] animals. The reduction in the number of PDGFRa[+ve] NG2-glia, MAG[+ve] mature oligodendrocytes and concomitant increase of APC[+ve] cells observed in NIMR *Sox3*[-/Y] mice (Figs 5B–5D and 6F–6G) was confirmed in FT *Sox3*[-/Y] ME but absent in Crick *Sox3*[-/Y] ME (Fig 6E–6F). Altogether, these data indicate that changes of gut microbiota affect oligodendrocyte lineage composition in ME, and occurrence of hypopituitarism in *Sox3*[-/Y] mutants.

In conclusion, because NG2-glia in the ME are the cells mostly affected by *Sox3* loss and their proliferation, but not differentiation to oligodendrocytes, is corrected by either low-dose Aspirin treatment or by a 'Crick-typical' profile of gut microbiota, we propose that sufficient numbers of NG2-glia are required in the ME to support pituitary cell maturation as the animal becomes independent at weaning.

## Discussion

Congenital pituitary hormone deficiencies, found with a frequency of about 1/4000 [50], can significantly affect quality of life. However, relatively few genes have been identified as being causative, with most of these having critical roles in prenatal pituitary development [50–52]. Moreover, ways to treat patients are limited to replacement of missing hormones, but in way that does not mimic their normal physiological release by the pituitary. Mutations affecting SOX3 are associated with panhypopituitarism in mice and humans [25,26], however the

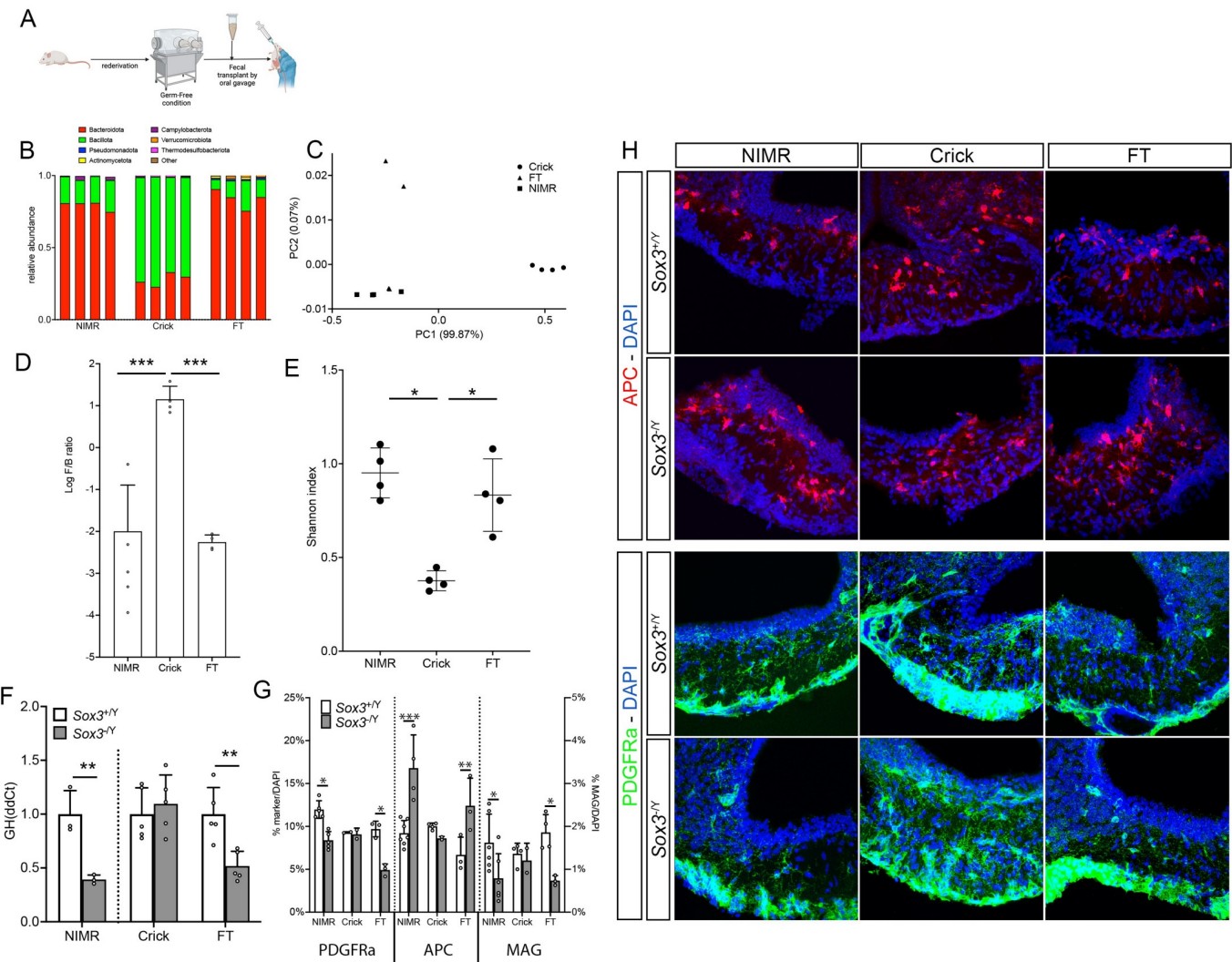

**Fig 6. *Sox3*<sup>-/Y</sup> NG2-glia ME composition and GH deficiencies is affected by changes of gut microbiota.** (A) Graphical representation of the process the fecal transplant was performed. Created with BioRender.com (B) Abundance of different bacteria phylum in gut microbiota of mice housed at NIMR, Crick or faecal transplanted (FT). (C) PCA analysis of gut microbiota of samples from Crick (circle), NIMR (square) and FT (triangle). (D) Log Ratio of Firmicutes/ Bacteroidetes (F/B) of mice housed at NIMR, Crick or FT. (E) Shannon index of species of mice housed at NIMR, Crick or FT. (F) Expression of *Gh* (ddCt) in 2-month NIMR, Crick and FT *Sox3*<sup>+/Y</sup> (white column) and *Sox3*<sup>-/Y</sup> (grey columns) pituitaries. (G) Composition of the median eminence in regard to the oligodendrocyte lineage using PDGFRa, APC and MAG of NIMR, Crick and FT *Sox3*<sup>+/Y</sup> (white columns) and *Sox3*<sup>-/Y</sup> (grey columns) mice. (H) Fluorescent immunolabelling for APC and PDGFRa on 2-month NIMR, Crick and FT *Sox3*<sup>+/Y</sup> and *Sox3*<sup>-/Y</sup> hypothalami. *: $p < 0.05$; **: $p < 0.01$; ***: $p < 0.001$.

etiology of this was unknown. Here, our results suggest that the phenotype has a hypothalamic origin, because conditional deletion of the gene using Nestin-Cre is sufficient to reproduce the hypopituitarism characterising the null phenotype. Nestin-cre is expressed in some locations outside the nervous system, such as by cells in the kidney and heart [53], but these have not been reported to be sites of *Sox3* expression, and while both Nestin and *Sox3* are expressed quite broadly in the nervous system, the crucial regulatory role of the hypothalamus on the pituitary makes it highly likely that it is defects in this region that underlie the hypopituitarism phenotype. Moreover, adding weight to this premise, we found that the onset of defects in the ME correlate with the failure of endocrine cell maturation in the pituitary after weaning, and this in turn explains why they can only make low levels of hormones.

The ME is a crucial structure conveying information from and to the brain, but how and when it becomes functional and controls pituitary output is not well characterized [54–58]. We find that SOX3 is required for self-renewal of both hypothalamic NSCs, which are part of the tanycyte population lining the third ventricle including the ME, and NG2-glia within the ME. SOX3 is also required for the differentiation of the latter to mature oligodendrocytes, however in two conditions where the panhypopituitarism is rescued, via the effects of aspirin or altered gut microbiota, although cell proliferation returns to normal, there is still a failure of oligodendrocyte differentiation suggesting that myelination of axons is not essential for ME function. Given that the hypothalamic NSCs generate very few neurons, if any, post-puberty (for review see [59]), it would seem most likely that the pituitary phenotype is due to insufficient numbers of glia. Indeed, our data point to the NG2-glia themselves as having a critical role in inducing or allowing pituitary endocrine cell maturation. It follows that there must be processes that occur around weaning in the ME, perhaps involving an increase in levels and/or alteration in patterns of hormone releasing factors secreted into the hypophyseal portal system, which are indispensable for maturation of the juvenile pituitary.

Weaning is a crucial transition period when the suckling pup becomes independent [60] and this is accompanied by both behavioural and physiological changes [61]. The sensitivity of this stage is highlighted by mutations in genes affecting appetite control, another crucial function of the hypothalamus [62], such as in *Neurogenin3* [63] and U11 spliceosomal RNA (*Rnu11*) [64], or affecting the maturation of endocrine cells in the pancreas [65]. It would of interest to ask if altering the time of weaning of $Sox3^{-/Y}$ mice has any influence on the onset of pituitary hormone deficiencies.

## SOX3 in hypothalamic neural stem cells

We show that tanycytes, a proportion of which are hypothalamic neural stem cells, divide less frequently in $Sox3^{-/Y}$ mice compared to $Sox3^{+/Y}$ animals. This is also reflected in neurosphere cultures derived from the hypothalamus, where those from $Sox3^{-/Y}$ mice post-weaning fail to be maintained for more than a few passages in contrast to cultures initiated at earlier stages or from other neurogenic niches at any stage. This is similar to the effects of the loss of *Sox2* in the adult dentate gyrus [66]. We showed that lineage commitment is not affected in differentiated $Sox3^{-/Y}$ hypothalamic neurospheres, consistent with a previous report indicating that over-expression of SOX3 does not promote neurogenesis in embryonic telencephalic NSC culture [67]. This is in contrast with SOX1 and SOX2, the other members of the SOXB1 family, which have essential roles in neuronal differentiation [66,68].

SOX3 is present in all adult NSC niches ([41] and this study), however, its loss exclusively affects hypothalamic stem/progenitor cells post puberty. In the embryonic brain, SOX3 is expressed at its highest levels in the infundibulum which will give rise to the ME. Although the related factor SOX2, with which it is known to show functional redundancy, is expressed throughout the neuroepithelium, the absence of SOX3 does have consequences at this stage with respect to induction of the pituitary and its morphological development [26]. It seems likely that the hypothalamic stem cells, in which SOX2 is also expressed, are similarly more sensitive to the loss of SOX3.

## Sox3 and ME NG2-glia

NG2-glia are the most clearly affected cell type in the $Sox3^{-/Y}$ ME, with both their proliferation and differentiation being disrupted, which suggests that deficits in this population are responsible for the hypopituitarism. We cannot exclude an additional role for tanycytes, whose maintenance is also affected in *Sox3* mutants, however, hypothalamic stem cells do not normally

give rise to NG2-glia in adult mice [69]; it is therefore unlikely that the reduction in the number of ME NG2-glia in *Sox3* mutants is secondary to the proliferation defect of the stem cells. Tancytes are directly involved in regulating Gonadotropin hormone-releasing hormone and Thyrotropin-releasing hormone release into the blood stream at the ME level ([70,71] and for review: [8]) and it is possible that SOX3 plays a role in this, as well as in their self-renewal as we reveal by proliferation and neurosphere assays. However, unless other neurohormones are similarly affected this would not explain the pan-hypopituitarism we see in Sox3 mutants, moreover, the phenotype and its rescue correlate with alterations in NG2 glia. SOX3 function in ME NG2-glia could relate to its role in the spinal cord where it is required for their differentiation [24]. ME NG2-glia proliferate faster than their parenchymal counterparts [5], perhaps because they are exposed to systemic blood circulation. This higher cell-turnover suggests that modest defects affecting this population may be more pronounced in the ME explaining why the defects seen in $Sox3^{-/Y}$ mice are restricted to this domain.

Alternatively, rather than just being precursors of oligodendrocytes, NG2-glia have other roles that involve crosstalk with neurons, astrocytes and microglia (for review, see [7]), all of which are found within the ME [6]. Indeed, a direct role for NG2-glia in the ME for neuronal function has already been indicated, whereby their ablation is associated with obesity attributed to degeneration of the dendritic processes of arcuate nucleus leptin receptor neurons [4]. While we do not yet know if NG2-glia are required for pituitary hypophysiotropic neuron integrity, it is worth noting that hypopituitarism occurs in both humans and rats after cranial irradiation [72,73] where NG2-glia may be the most sensitive cell type given their high rate of proliferation compared to other cell types in the brain.

## Aspirin restores pituitary function in Sox3 mutants

A further indication that the hypopituitarism seen in the *Sox3* mutants is likely due to a deficiency of NG2-glia comes from the restoration of both pituitary function and proliferation of NG2-glia with low dose aspirin, while differentiation into mature oligodendrocytes is still impaired. The canonical mode of action of low dose aspirin involves the suppression of cyclo-oxygenase (COX) 1 activities [74]. COX1 is expressed in microglia [75] and tanycytes [76] and its inhibition in these could affect NG2-glia. However, it was shown recently that targeted ablation of microglia led to a decrease in NG2-glia, differentiating and mature oligodendrocytes in the ME [5] and indicated a relationship between microglia and myelin turnover. Detailed analyses would be needed to determine whether inhibition of COX1 activities in microglia by aspirin is responsible for the restoration of NG2-glia proliferation in *Sox3* mutants or if COX-independent mechanisms are also relevant [77]. In contrast, low dose aspirin increases differentiation rather than proliferation of cultured NG2-glia by inhibiting Wnt signalling [27], but the situation may be different *in vivo*, where the effects of aspirin might also be mediated by other cells within or outside the ME. COX signalling is active in tanycytes. Furthermore, inhibition of the pathway at the ME has been proposed to impair the ovarian cycle through its effect on tanycytes [76]. Therefore, we cannot exclude a tanycytic component of the beneficial effect that low-dose aspirin has on Sox3-/Y hypopituitarism. However, dissecting the relative contribution of tanycytes, NG2-glia and microglia to the phenotype will require new methods that avoid the use of tamoxifen, or of other agents that can themselves affect pituitary function.

## Gut microbiota and hypopituitarism in Sox3 mutant mice

We found that changes in microbiome composition can alter both the NG2-glia phenotype and development of hypopituitarism in *Sox3* mutants. Communication between the gut microbiota and the brain could be via chemical compounds released into the blood stream,

neuronal signalling, or by cells of the immune system [78]. For example, changes on the gut microbiome affect circulating levels of the promyelinating factor IGF1, which in turn affects cortical myelin basic protein but not NG2 expression [79,80]. Moreover, it is now thought that an intact gut microbiota is necessary for a proper response from the hypothalamo-pituitary-adrenal axis. Indeed, germ-free mice, which have a minimal gut microbiota, display an exacerbated response to early life stress [81]. It may be noteworthy that gut microbiome composition changes drastically during early postnatal life, and only stabilises after weaning, which is when the absence of SOX3 becomes relevant [82]. Understanding how a post-weaning stabilised microbiota of a particular composition could rescue both NG2-glia in the ME and pituitary deficiencies will be important especially if this can be shown to be relevant in humans.

In this respect, a 'weaning reaction' to microbiota has been described, and while this was associated with development of the immune system and protection against immunopathologies [83], gut bacterial cell wall components, muropeptides, can also be sensed directly via Nod1 and Nod2 receptors on cells in the brain, including the hypothalamus [84]. It is worth noting that the predominant phylum of gut microbiota at NIMR were Gram negative Bacteroidetes, which possess a unique diaminopimelate-containing GlcNAc-MurNAc tripeptide muropeptide (GM-TriDAP) sensed by NOD1, while the most abundant phyla at the Crick are Gram-positive Firmicutes carrying the GlcNAc-MurNAc dipeptide (GM-Di), which can be sensed by both NOD1 and NOD2 [85–87]. It will be important to determine which cells in the ME are expressing NOD1 and NOD2, especially post-weaning, and how these may respond to the different muropeptides. This might provide an explanation for the sensitivity of the *Sox3* pituitary phenotype to gut microbiota. To confirm the contribution of the microbiome to the Sox3 null phenotype, experiments would have to be performed in more rigorous settings in which mice under germ-free conditions would be faecal transplanted with either NIMR or Crick materials. While we have not formally tested the effect of the different diets on the microbiome, GH-deficient FT mice and their progeny have been kept on the Crick diet suggesting that the phenotypic effect is primarily due to microbiome differences. Furthermore, certain bacteria within the gut microbiota could trigger inflammation throughout the body (for review [88]). Further experiments need to be undertaken to determine the level of inflammation in the different settings and whether the beneficial effect of aspirin is due to its anti-inflammatory action.

## Conclusions

The data presented here strongly suggest that NG2-glia in the ME, and the presence of SOX3 in these, are crucial to forming a functional hypothalamo-pituitary axis post-weaning. Moreover, while hypopituitarism seen in both mice and humans lacking SOX3 can be a robust phenotype, we find that it can be modified by external influences, namely by low dose aspirin or gut microbiota. Consequently, because of the potential link between reduced NG2-glia after irradiation and hypopituitarism, it is of interest to explore whether low dose aspirin treatment and/or changes in microbiota composition could have rescuing effects in this context too.

## Materials and methods

### Ethics statement

All experiments carried out on mice were reviewed by the NIMR and Crick Animal Welfare and Ethical Review Committees and approved under the UK Animal (scientific procedures) Act (approved Project licence 80/2405, 70/8560 and PP8826065).

## Mice

Mice carrying the *Sox3*-conditional allele (*Sox3*<sup>floxGFP/Y</sup>–MGI:5550577) and *Sox3* null allele, where the *Sox3* open reading frame is replaced by that of *gfp*, (*Sox3*<sup>ΔGFP/Y</sup> referred as *Sox3*<sup>-/Y</sup>– MGI: 3036671) [26], and alleles for *Pou1f1*-Cre [31], *Nestin*-Cre (MGI: 2176222) [30], *Pdgfra*-CreERT2 (MGI: 3832569) [89], *Rosa26*<sup>ReYFP</sup> (MGI: 2449038) [90] are maintained on MF1 background (random bred, NIMR:MF1 or HsdOla:MF1 for Crick and FT *Sox3*<sup>-/Y</sup>) and genotyped as described or by Transnetyx. Mice had access to food and water ad libitum with diet 5021 (Labdiet) at NIMR and diet 2018S (Envigo) at the Crick.

## Radioimmunoassay

Pituitaries were homogenized in phosphate-buffered saline and anterior pituitary hormonal contents measured by radioimmunoassay (RIA) [91] using National Hormone and Pituitary Program reagents kindly provided by A.L. Parlow.

## Neurosphere forming assay and monolayer culture

Hypothalamus and sub-ependymal zone (SEZ) of the lateral ventricles were dissected and cultivated in proliferation medium as described previously [37,92]. Neurospheres were passaged using TrypLE express (Invitrogen). For monolayer culture, coverslips were coated with reduced growth factor matrigel (BD biosciences) and cells were plated at $2.10^4$cells/ml.

## Substance administration, sample processing and histology

BrdU (Sigma) and Tamoxifen (Sigma) were administered intraperitoneally at 100μg/g and 0.2mg/g body weight respectively twice daily for BrdU, and daily for Tamoxifen for 5 consecutive days. Aspirin pellets (12mg/day/kg for 21 days, 7.5mg per pellet—Innovative Research of America) were administered subcutaneously. Adult animals were perfused intracardially with chilled 4% paraformaldehyde, brains were embedded in 2.5% agarose and free-floating sections obtained using a vibratome (Leica). Neurospheres or cell monolayers were fixed with chilled 4% paraformaldehyde for 20 minutes on ice. For histology, samples were fixed in Bouin's solution, processed for wax embedding, sectioned, and stained with haematoxylin and eosin. For osmolality measurement, urine was collected in ependorf tubes and osmolality was measured with a vapour pressure osmometer.

## Immunofluorescence, in situ hybridisation, TUNEL assay and qPCR experiments

Immunofluorescence was performed on 50μm free-floating sections, neurospheres or cell monolayers plated on coverslips. Samples were incubated in blocking solution (PBS, 10% donkey serum, 0.1% Triton X-100) and primary antibodies were incubated overnight at 4˚C (Table 1). Detection was performed using a 1:500 dilution of anti-rabbit, anti-mouse or anti-goat secondary antibodies conjugated to Alexa-488, Alexa-555, Alexa-594 or Alexa-647 (Molecular probes). DNA was labelled using DAPI and sections were mounted in Aquapoly-mount (polysciences). RNA *in situ* hybridization on free-floating sections was performed as previously described [93] using *Rax* [94], *Ghrh*, *somatostatin and arginine vasopression (AVP)* probes (gift from JP. Martinez-Barbera, ICH, UCL, UK). TUNEL experiments were carried out following the supplier manual (ApopTag Kit, Millipore). For gene expression, RNA extraction was carried out following the supplier manual (miRNeasy kit, Qiagen). Complementary DNA was synthetised using 1x qScript cDNA SuperMix (Quantabio). Quantitative PCR (qPCR) were performed using PowerUp SYBR green (ThermoFisher Scientific) with primers for *Gh*

**Table 1. List of primary antibodies used.**

| Antigen | Host | dilution | Vendor |
|---|---|---|---|
| βIII-tubulin | Rabbit | 1/100 | Abcam |
| APC | Mouse | 1/200 | Merck |
| BrdU | Mouse | 1/500 | GE life sciences |
| BrdU | Rat | 1/400 | Abcam |
| Nestin | Mouse | 1/100 | DSHB |
| GFAP | Mouse | 1/1000 | Sigma |
| GFAP | Rabbit | 1/1000 | Sigma |
| GFP | Goat | 1/200 | Abcam |
| GFP | Rabbit | 1/200 | Invitrogen |
| MAG | Rabbit | 1/200 | NEB |
| NeuN | Mouse | 1/200 | Millipore |
| NG2 | Rabbit | 1/200 | Chemicon |
| Olig2 | Rabbit | 1/500 | 2b scientific |
| PDGFRa | Goat | 1/100 | R&D systems |
| PDGFRa | Rat | 1/500 | BD Pharmingen |
| Phospho-histone H3 | Mouse | 1/500 | Abcam |
| Pit-1 | Rabbit | 1/1000 | gift from Dr Rhodes, Indiana University, USA |
| Sox2 | Rabbit | 1/300 | Millipore |
| Sox2 | Goat | 1/300 | ISL |
| Sox3 | Goat | 1/100 | R&D systems |
| Sox9 | Goat | 1/300 | R&D systems |
| SoxE | Rabbit | 1/1000 | - |
| TSH | Rabbit | 1/100 | A.L. Parlow |

5'TGGGCAGATCCTCAAGCAAACCTA3' and 5'GAAGGCACAGCTGCTTTCCACAAA3'. Glyceraldehyde 3-phosphate dehydrogenase (GAPDH) expression was used as house-keeping gene (5'TTCACCACCATGGAGAAGGC3' and 5'CCCTTTTGGCTCCACCCT3'). Each biological sample was run in technical triplicates and, at least, three biological samples were analysed. Relative expression of gene of interest was calculated according to the ddCt method [95] and represented as fold change to the corresponding control.

## Faecal transplant and gut microbiota sequencing

Crick mice were rederived via embryo transfer into germ-free recipient females and maintained in isolators under germ-free conditions at Taconic Biosciences facility (Taconic Biosciences). Frozen NIMR faecal materials were homogenised in 100ul PBS at room temperature for 1 minute. Freshly prepared supernatant (around 75 – 80ul) was then administered by oral gavage once per day for 2 days (TruBIOME, Taconic). Fecal transplanted (FT) mice were sent to the Crick facility for further breeding and maintenance. The FT mice were kept on Crick diet. For sequencing, freshly collected or snap-frozen faecal materials collected between 10am and 12-noon (3 to 5h in the light cycle) from 4 different animals in each group were incubated in stabilisation buffer and subjected to shotgun whole genome sequencing with read depth of 2 millions paired-end reads (Transnetyx). Data was analysed with OneCodex online software. Gut microbiota datasets are publicly available at the Sequence Read Archive (SRA) of the National Library of Medicine (PRJNA1113545).

## Quantification and statistical analysis

Images were captured by confocal microscopy (Leica SP5 or SPE) or by light microscope (Leica DMRA2) and analysed using Leica LAS AF, ImageJ, OpenLab or Volocity softwares. For anterior pituitary images, cell density was measured by counting the number of haematoxylin positive nuclei per 35mm$^2$ and pituitary volume was measured using ImageJ. TSH$^{+ve}$ cell volume was measured using Volocity. Immunostained *Sox3*$^{+/\Delta GFP}$ neurospheres are imaged and quantified according to the number of SOX3 and GFP positive cells they contain. The spheres are categorized into mainly SOX3$^{+ve}$ spheres or mainly GFP$^{+ve}$ spheres. Experiments are carried out at least in triplicate and at least three different fields were quantified for each experiment. For BrdU quantification, only BrdU$^{+ve}$ cells 100μm either side of the 3$^{rd}$ ventricle and within the median eminence were counted. α and β tanycytes were determined by their relative location within the hypothalamus. For statistical calculation, angular transformation was performed on percentage values. Unpaired t-test with Welch's correction and two-way ANOVA were used for the analysis of statistical significance and error bars represent the standard deviation.

## Supporting information

**S1 Fig. Pituitary expression of SOX3 and RIA.** (A) Fluorescent immunolabelling for SOX3, POU1F1 and Prolactin (PRL) on 2-month old *Sox3*$^{+/Y}$ anterior pituitaries. Scale bars represent 10μm. (B) PRL contents in 2-month *Sox3*$^{floxGFP/Y}$ and *Pou1f1*-Cre; *Sox3*$^{floxGFP/Y}$. (C) PRL, TSH and LH contents in 4 weeks and 5 weeks *Sox3*$^{+/Y}$ and *Sox3*$^{-/Y}$ pituitaries. Concentrations on the Y axis: LH, 10$^{-6}$g per pituitary; PRL, 10$^{-6}$g per pituitary; TSH, 10$^{-7}$g per pituitary. *: p<0.05.
(TIF)

**S2 Fig. Median eminence expression of SOX3.** (A) Percentage of SOX3+ve cells in *Sox3*$^{+/Y}$ hypothalami expressing NeuN, PDGFRa or GFAP. (B-C) Fluorescent immunolabelling for SOX3, NG2 (B) and MAG (C) on 2-month *Sox3*$^{+/Y}$ MEs. SOX3 is expressed in NG2$^{+ve}$ NG2-glia but not in MAG$^{+ve}$ mature oligodendrocytes. Scale bars represent 10μm.
(TIF)

**S3 Fig. SOX3 expression in *Sox3*$^{-/Y}$ hypothalamus and neurospheres.** (A) Fluorescent immunolabelling for SOX3 on 2-month old *Sox3*$^{-/Y}$ hypothalamic. (B) Fluorescent immunolabelling for SOX3 and Nestin on *Sox3*$^{+/Y}$ and *Sox3*$^{-/Y}$ neurospheres. Note the absence of SOX3 in *Sox3*$^{-/Y}$ spheres.
(TIF)

**S4 Fig. Post-weaning apoptosis analysis *in vitro* and *in vivo*.** (A) Percentage of TUNEL$^{+ve}$ cells in *Sox3*$^{+/Y}$ (white columns) and *Sox3*$^{-/Y}$ (grey columns) of monolayer cultures derived from adult animals at secondary, tertiary, and quaternary passages. No significant difference is observed. (B) TUNEL staining on *Sox3*$^{+/Y}$and *Sox3*$^{-/Y}$ 2-month hypothalami. Arrows point to TUNEL$^{+ve}$ cells. Very few positive cells are present in both samples. Scale bar represents 100μm.
(TIF)

**S5 Fig. Pre-weaning hypothalamic neural/progenitor cells analyses *in vitro* and *in vivo*.** (A) Total number of cells counted after each passage from hypothalamic derived (red/pink) or SEZ derived (dark and light blue) NSCs from ten-day old *Sox3*$^{+/Y}$ (dark blue/red) and *Sox3*$^{-/Y}$ mice (light blue/pink). *Sox3*$^{-/Y}$ NSCs from pups self-renew normally. (B) Percentage of phospho-histone H3$^{+ve}$ cells in secondary progenitor monolayer cultures originated from ten-day

old $Sox3^{+/Y}$ and $Sox3^{-/Y}$ hypothalami. $Sox3^{-/Y}$ progenitors proliferate normally. (C) Percentage of neurospheres containing mostly SOX3$^{+ve}$ cells (white columns) or mostly GFP$^{+ve}$ cells (grey columns) derived from ten-day old $Sox3^{+/\Delta GFP}$ SEZ or hypothalami. SOX3$^{-ve}$, GFP$^{+ve}$ progenitors are maintained in these cultures from young females. (D) Number of BrdU$^{+ve}$ cells in one-week $Sox3^{+/Y}$ (white columns) and $Sox3^{-/Y}$ (grey columns) hypothalamic parenchymas and MEs. The number of BrdU$^{+ve}$ cells is similar in $Sox3^{-/Y}$ compared to $Sox3^{+/Y}$ samples. (TIF)

**S6 Fig. Hypothalamic neurosphere differentiation potential.** (A-C) Fluorescent immunolabelling for GFAP (astrocytes), βIII-tubulin (neurons), CNPase (oligodendrocytes) on 6-weeks (A) and 1-week (C) $Sox3^{+/Y}$ hypothalamic differentiated progenitor monolayers. (B) Percentage of cells differentiated into βIII-tubulin$^{+ve}$ neurons and CNPase$^{+ve}$ oligodendrocytes derived from 6-weeks $Sox3^{+/Y}$ (white columns) or $Sox3^{-/Y}$ (grey columns) hypothalami. Scale bars represent 30μm. (TIF)

**S7 Fig. AVP axis is not affected in *Sox3* mutants.** (A) *In situ* hybridization for *AVP* on 2-month $Sox3^{+/Y}$ and $Sox3^{-/Y}$ hypothalami. (B) $Sox3^{+/Y}$ and $Sox3^{-/Y}$ posterior pituitary sizes at different post-natal weeks. $Sox3^{-/Y}$ posterior pituitaries are not smaller to that of $Sox3^{+/Y}$ animals. (C) Urine osmolality measurement at different post-natal weeks. (TIF)

**S8 Fig. Low dose aspirin treated *Sox3*$^{-/Y}$ myelin analyses and RIA.** (A) Transmission electron microscopy images of 2-month $Sox3^{+/Y}$, $Sox3^{-/Y}$ with placebo and aspirin-treated $Sox3^{-/Y}$ MEs. Myelinated axons are missing in aspirin-treated $Sox3^{-/Y}$ MEs. (B) ACTH contents in 2-month $Sox3^{+/Y}$, $Sox3^{+/Y}$ with low dose aspirin, $Sox3^{-/Y}$ with low dose aspirin and $Sox3^{-/Y}$ with placebo pituitaries. Note the rescue in ACTH contents in $Sox3^{-/Y}$ mice receiving low dose aspirin. **: p<0.01. (TIF)

**S9 Fig. Diet composition and gut microbiota phyla composition.** (A) Composition of rodent chow at NIMR and Crick. (B) Abundance of different bacteria phylum in gut microbiota of mice housed at NIMR, Crick or faecal transplanted (FT) focussing only on lowly expression phyla. (TIF)

## Acknowledgments

We are grateful for the help and support of all recent past and present members of the Lovell-Badge lab. We are indebted to Dr. AF Parlow and the NHPP for reagents for the RIA and immunofluorescence experiments. We thank Simon Rhodes (Indiana University School of Medicine) for the generous gift of POU1F1 antibodies, J.P. Martinez-Barbera (University College London, UK) for *Ghrh* and *Sst* probes and S. Blackshaw for *Rax* probe. We thank Patrice Mollard (IGF, Montpellier, France) foir helpful discussions. We also thank Biological Services, the light microscopy, the experimental histopathology and the electron microscopy platforms at the MRC National Institute for Medical Research and the Francis Crick Institute, for their excellent assistance and technical support and BioRender for graphical representation.

## Author Contributions

**Conceptualization:** Robin Lovell-Badge.

**Data curation:** Christophe Galichet.

**Formal analysis:** Christophe Galichet.

**Writing – original draft:** Christophe Galichet, Karine Rizzoti.

**Writing – review & editing:** Christophe Galichet, Karine Rizzoti, Robin Lovell-Badge.

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
