## [Decision Letter · Decision Letter 0]

24 Apr 2024

Dear Dr Galichet,

Thank you very much for submitting your Research Article entitled 'Sox3-null hypopituitarism depends on median eminence NG2-glia and is influenced by aspirin and gut microbiota' to PLOS Genetics.

The manuscript was fully evaluated at the editorial level and by independent peer reviewers. The reviewers appreciated the attention to an important problem, but raised some substantial concerns about the current manuscript. Based on the reviews, we will not be able to accept this version of the manuscript, but we would be willing to review a much-revised version. We cannot, of course, promise publication at that time.

If you decide to revise the manuscript for further consideration at PLOS Genetics, please aim to resubmit within the next 60 days, unless it will take extra time to address the concerns of the reviewers, in which case we would appreciate an expected resubmission date by email to plosgenetics@plos.org.

We are sorry that we cannot be more positive about your manuscript at this stage. Please do not hesitate to contact us if you have any concerns or questions.

Yours sincerely,

Bruce A. Hamilton

Academic Editor

PLOS Genetics

Hua Tang

Section Editor

PLOS Genetics

Reviewer's Responses to Questions

**Comments to the Authors:**

Reviewer #1: Comments on Galichet et al., “Sox3-null hypopituitarism depends on median eminence NG2-glia and is influenced by aspirin and gut microbiota”:

• The authors establish a correlation between hypopituitarism and NG2+ cell alterations in the median eminence. However, a causal relationship is not definitively demonstrated. The interpretation, as well as the title ("depends on"), should be adjusted to reflect this limitation.

• The focus on NG2 glia as the primary mechanism for hypopituitarism may need further justification. Since Sox3 deletion affects the self-renewal of hypothalamic NSCs, particularly tanycytes, another explanation could be an alteration in the tanycyte population's ability to regulate hypothalamic neurohormone release in the ME vasculature.

• Consider the role of the Cox-prostaglandin pathway in tanycytic control of neurohormone release, especially given aspirin's effect on NG2-glia. This pathway might contribute to the beneficial effects of aspirin on pituitary function.

• Provide evidence for the loss of Sox3 expression in Sox3-null mice, both in vivo and in vitro.

• Further characterization of pituitary defects could include evaluating pituitary hormone gene expression via in situ hybridization or RT-qPCR.

• Quantify Sox3-expressing cells (line 206-210) to enhance characterization.

• Include illustrative microphotographs for Figure 1F and corresponding quantification for Figure 1G.

• In Figure 2G, the statement about Rax expression needs adjustment due to apparent differences in microphotographs. Ensure consistency between images and conclusions.

• For Figure 3C, where BrdU incorporation in tanycytes is quantified, clarify the method of tanycyte identification and provide an illustrative microphotograph. Indicate any statistical differences on the bar graph.

• In Figure 3F, the upper panel's Sox2 green signal appears non-specific. This should be addressed for clarity.

Minor comments:

• Specify that hypothalamic NSCs mentioned (line 132, 211, 247) correspond to tanycytes for clarity.

• Clarify if "SEZ" refers to the SEZ of the lateral ventricles.

• Early mention of the Sox3 gene location on the X-chromosome (line 239) would clarify the naming convention for transgenic males ("Sox3-/Y").

• In Figure 1D, indicate the location of the anterior lobe on the microphotographs for clarity.

Reviewer #2: Please note that the editors have requested that we evaluate the microbiome components of this manuscript that investigates the neurological mechanisms underlying hypopituitarism. Key findings include the influence of external factors like low-dose aspirin and alterations in gut microbiota on reversing some of the effects of Sox3 deletion, such as promoting the proliferation of NG2-glia and mitigating hypopituitarism. These results suggest that managing gut microbiota could be a novel therapeutic approach to treating or managing hypopituitarism, and contributes to a growing body of research on gut-brain axis, illustrating how gut microbiota can influence brain health and disease. The study not only broadens our understanding of hypopituitarism but also opens potential therapeutic pathways involving dietary modifications or microbiome management. Of note, the microbiome components of the study do not provide mechanistic insight, but that is not in the scope of this manuscript. Nevertheless, the microbiome components of the study could be made more informative and these observations can be better described to help with future research in the field. Overall, the authors are to be commended for rigorously investigating a change in their phenotype once they changed vivariums and that their data suggests that this change could be driven by microbiome changes.

Major Comments:

The figures/figure panels do not seem to be presented in order throughout the manuscript which is potentially confusing for readers. We highly recommend presenting the figures/figure panels in order to avoid confusion.

It may be a good idea to include a figure panel with a schematic of how the FT experiment was performed. It would be helpful to more immediately know if this was a single fecal transplant or multiple doses of the bacteria.

It is disappointing that the FT experiment only included the rederivation of the NIMR microbiome in germ-free mice in the Crick facility. Ideally this should also included transplant of Crick microbiome in germ-free mice as a control as well. Importantly, what type of diet was administered to germ-free mice? The Crick diet or the NIMR diet? Far more methodological clarity is necessary here to truly determine whether this is microbiome driven. I would highly recommend repeating the experiment with these appropriate controls to delineate the true contribution of microbiome to their phenotype. Otherwise, the authors should mention that this is a severe limitation of their study and how further more rigorous testing is required to determine the contribution of the microbiome vs. diet.

For Panel A in Fig 6, the smallest sections of the pie charts are difficult to identify. It looks like Firmicutes (green) appear twice. Improving the image quality will enhance the clarity of this panel. Overall, we find pie charts to be a poor method for conveying compositional differences. Alternatively, the authors could use a stacked taxonomic barplot to display these data, which would make it easier to compare across groups. In addition, to convey individual data, we recommend that the stacked bar plots to be for individual mice with the different groups clustered together. Importantly, for a more holistic view of the gut microbiota composition differences, ordination plots like PCA or PCoA should be used to visualize the overall similarity between microbial communities from different groups, with statistical differences conveyed with PERMANOVA. As it stands the description of composition differences, and whether they are truly statistically different is quite incomplete.

For Panel B in Fig 6, it is now recognized that firmicutes/bacteroidetes ratio is primarily driven by the amount of fat in the mouse diet, whereas earlier microbiome studies seem to imply that there was something more significant about this particular phylum ratio. Though the authors are illustrating compositional differences between their cohorts, it may be a good idea to use log-ratios to convey this information which is a more accurate way of depicting this data and convey a bit more information about compositional differences (see PMID 31222023, which does a better job of explaining the importance of log-ratios).

For Panel C in Fig 6, could you please provide additional details on how alpha diversity was measured in your study? Specifically, it would be helpful to know which alpha diversity metric(s) you employed (e.g., Shannon index, Simpson index, Chao1, observed species) and any software or analytical tools used to calculate these metrics.

The discussion provides a good overview of potential pathways through which the gut microbiota might influence brain function and development. However, further detailing the specific mechanisms, perhaps by elaborating on how these pathways specifically interact with NG2-glia or affect hypopituitarism, would deepen the reader's comprehension of the underlying biology.

The “fecal transplant and gut microbiota sequencing” under the Materials and methods section needs more elaboration:

Specify the process used for rederiving Crick mice into germ-free conditions. Include any specific procedures, conditions, or equipment used to ensure the mice were maintained germ-free.

Detail the method used for homogenizing NIMR fecal materials. Mention the volume of PBS used and any specific conditions (e.g., temperature, time) for homogenization.

Clarify whether the oral gavage was performed once or if it was part of a repeated schedule. Also, indicate the volume administered to each mouse.

Clarify length of treatment prior to phenotyping.

Specify the whole-genome sequencing process, including the sequencing platform used, the coverage depth aimed for, and any specific library preparation protocols followed.

Lines 411-413: The authors note the absence of hypopituitarism in Sox3-null mice after relocating to the Crick Institute. However, it is unclear whether the mice were directly transported from the previous facility, or were they part of a new breeding cohort established at Crick? If the former, could the authors specify the timeframe within which these changes manifested? Understanding the temporal dynamics of how dietary and environmental modifications impact gut microbiota composition—and subsequently, hypothalamic functions—is crucial for deepening our insights into the interplay between diet, gut microbiota, and host physiological responses.

There needs to be a statement in the text about how many mice from each group were included in this study. Based on graphs it appears that 4-5 mice were used. If so, this should be made explicit in the methods. Also if any mice were excluded from analysis, this should be noted as well. Additionally in lines 418 - 419, the authors should state the number of mice whose gut microbiota was compared from the previous and current institution.

Minor Comments:

There are instances where "SOX3" is used when the discussion seems to be specifically about mice. For example in line 105. In these cases, for strict adherence to naming conventions, it would be clearer to use "Sox3" when referring to findings specifically in mice. This nuanced approach to naming helps clarify which species' gene is being discussed, providing clear differentiation between human and mouse research findings. It's a subtle distinction but important for scientific accuracy and clarity.

In line 139, the abbreviation 'GH' is used without prior definition. The full term 'growth hormone (GH),' provided in the following line, should be introduced at its first occurrence in line 139. Similarly on line 152, PRL, TSH and LH are introduced without prior definition. Additionally on line 153, GHRH is introduced for the first time without prior definition.

The authors use “Sox3+/Y” or “controls” interchangeably, and “Sox3-/Y” or “Sox3-null” interchangeably throughout the manuscript. To improve clarity for readers, I suggest being consistent with one term.

**Have all data underlying the figures and results presented in the manuscript been provided?**

Reviewer #1: Yes

Reviewer #2: **No: **microbiome data

PLOS authors have the option to publish the peer review history of their article (what does this mean?). If published, this will include your full peer review and any attached files.

Reviewer #1: No

Reviewer #2: No

---

## [Decision Letter · Decision Letter 1]

23 Jul 2024

Dear Dr Galichet,

Thank you very much for submitting your revised Research Article entitled 'Hypopituitarism in Sox3 null mutants correlates with altered NG2-glia in the median eminence and is influenced by aspirin and gut microbiota' to PLOS Genetics.

The reviewers and editors appreciated the detailed and positive responses to previous reviews. Before acceptance, however, we would like you to respond to two minor points in the Materials and Methods raised by Reviewer 2: (1) while we appreciate that the commercial diets are detailed in supplemental materials, please note the vendor and product number in the text and (2) note the time of day used for stool collection and state whether these were identical, similar, or divergent between experiment arms.

In addition we ask that you upload a Striking Image with a corresponding caption to accompany your manuscript if one is available (either a new image or an existing one from within your manuscript). If this image is judged to be suitable, it may be featured on our website. Images should ideally be high resolution, eye-catching, single panel square images. For examples, please browse our archive. If your image is from someone other than yourself, please ensure that the artist has read and agreed to the terms and conditions of the Creative Commons Attribution License. Note: we cannot publish copyrighted images.

To resubmit, log into your Editorial Manager account and select the option 'Revise Submission' in the 'Submissions Needing Revision' folder.

Yours sincerely,

Bruce A. Hamilton

Academic Editor

PLOS Genetics

Hua Tang

Section Editor

PLOS Genetics

Reviewer's Responses to Questions

**Comments to the Authors:**

Reviewer #1: The authors satisfactorily answered Reviewer's comments

Reviewer #2: I would like to thank the authors for addressing the microbiome aspects of their experiments, especially with the reanalysis of results, the more detailed explanation of their methodology, and the discussion of limitations. All of these changes have improved the manuscript dramatically Although the FT experiment lacks some valuable controls, the authors' acknowledgement of this limitation in the discussion improves transparency, especially in this paper where the microbiome findings are quite interesting and quite incidental.

I would encourage the authors to mention the diet in their description of the experiment in the results. That is, in lines 434 and onwards: “To test whether changes in gut microbiota underlie rescue of hormonal deficiencies, we performed faecal transplant (FT) from NIMR animals into Crick animals ***, while maintaining the mice on the Crick diet.***” This will help readers understand which variables/aspects of the experiment are being manipulated.

Given a recent paper that states that timing of sample collection could affect microbiome results (PMID 38951660), I would like the authors to confirm that the time of sample collection at NIMR and Crick facility was the same/similar and include additional information about stool collection time to confirm that this is not an artifact of changes in stool collection.

**Have all data underlying the figures and results presented in the manuscript been provided?**

Reviewer #1: Yes

Reviewer #2: Yes

PLOS authors have the option to publish the peer review history of their article (what does this mean?). If published, this will include your full peer review and any attached files.

Reviewer #1: No

Reviewer #2: No

---

## [Editor Report · Decision Letter 2]

13 Aug 2024

Dear Dr Galichet,

We are pleased to inform you that your manuscript entitled "Hypopituitarism in Sox3 null mutants correlates with altered NG2-glia in the median eminence and is influenced by aspirin and gut microbiota" has been editorially accepted for publication in PLOS Genetics. Congratulations!

Yours sincerely,

Bruce A. Hamilton

Academic Editor

PLOS Genetics

Hua Tang

Section Editor

PLOS Genetics

Comments from the reviewers (if applicable):

**Data Deposition**

http://datadryad.org/submit?journalID=pgenetics&manu=PGENETICS-D-24-00062R2

**Press Queries**

---

## [Editor Report · Acceptance letter]

30 Aug 2024

PGENETICS-D-24-00062R2 

Hypopituitarism in Sox3 null mutants correlates with altered NG2-glia in the median eminence and is influenced by aspirin and gut microbiota 

Dear Dr Galichet, 

We are pleased to inform you that your manuscript entitled "Hypopituitarism in Sox3 null mutants correlates with altered NG2-glia in the median eminence and is influenced by aspirin and gut microbiota" has been formally accepted for publication in PLOS Genetics! Your manuscript is now with our production department and you will be notified of the publication date in due course.

With kind regards,

Anita Estes

PLOS Genetics

On behalf of:
